# Whole-Genome Analysis of *Acinetobacter baumannii* Strain AB43 Containing a Type I-Fb CRISPR-Cas System: Insights into the Relationship with Drug Resistance

**DOI:** 10.3390/molecules27175665

**Published:** 2022-09-02

**Authors:** Tingting Guo, Jie Yang, Xiaoli Sun, Yuhang Wang, Liying Yang, Guimei Kong, Hongmei Jiao, Guangyu Bao, Guocai Li

**Affiliations:** 1Department of Microbiology, School of Medicine, Yangzhou University, Yangzhou 225001, China; 2Jiangsu Key Laboratory of Zoonosis/Jiangsu Co-Innovation Center for Prevention and Control of Important Animal Infectious Diseases and Zoonoses, Yangzhou University, Yangzhou 225009, China; 3Jiangsu Key Laboratory of Integrated Traditional Chinese and Western Medicine for Prevention and Treatment of Senile Diseases, Yangzhou 225001, China; 4Department of Diagnosis, Affiliated Hospital of Yangzhou University, Yangzhou 225001, China

**Keywords:** *Acinetobacter baumannii*, CRISPR-Cas, whole-genome sequencing, antimicrobial resistance

## Abstract

The CRISPR-Cas system is a bacterial and archaea adaptive immune system and is a newly recognized mechanism for controlling antibiotic resistance gene transfer. *Acinetobacter baumannii* (*A. baumannii*) is an important organism responsible for a variety of nosocomial infections. *A. baumannii* infections have become problematic worldwide because of the resistance of *A. baumannii* to multiple antibiotics. Thus, it is clinically significant to explore the relationship between the CRISPR-Cas system and drug resistance in *A. baumannii*. This study aimed to analyze the genomic characteristics of the *A. baumannii* strain AB3 containing the type I-Fb CRISPR-Cas system, which was isolated from a tertiary care hospital in China, and to investigate the relationship between the CRISPR-Cas system and antibiotic resistance in this strain. The whole-genome sequencing (WGS) of the AB43 strain was performed using Illumina and PacBio sequencing. The complete genome of AB43 consisted of a 3,854,806 bp chromosome and a 104,309 bp plasmid. The specific characteristics of the CRISPR-Cas system in AB43 are described as follows: (1) The strain AB43 carries a complete type I-Fb CRISPR-Cas system; (2) Homology analysis confirmed that the *cas* genes in AB43 share high sequence similarity with the same subtype *cas* genes; (3) A total of 28 of 105 *A. baumannii* AB43 CRISPR spacers matched genes in the bacteriophage genome database and the plasmid database, implying that the CRISPR-Cas system in AB43 provides immunity against invasive bacteriophage and plasmids; (4) None of the CRISPR spacers in *A. baumannii* AB43 were matched with antimicrobial resistance genes in the NCBI database. In addition, we analyzed the presence of antibiotic resistance genes and insertion sequences in the AB43 strain and found that the number of antibiotic resistance genes was not lower than in the “no CRISPR-Cas system” strain. This study supports the idea that the CRISPR-Cas system may inhibit drug-resistance gene expression via endogenous gene regulation, except to the published mechanism that the CRISPR-Cas system efficiently limits the acquisition of antibiotic resistance genes that make bacteria sensitive to antibiotics.

## 1. Introduction

*Acinetobacter baumannii* (*A. baumannii*) is an opportunistic Gram-negative pathogen that can survive in medical devices and on surfaces. Globally, *A. baumannii* infection is becoming increasingly problematic because of multiple antibiotic resistance, especially affecting patients in intensive care units and significantly limiting treatment options [1]. *A. baumannii* is one of the six ESKAPE pathogens (*Enterococcus faecium*, *Staphylococcus aureus*, *Klebsiella pneumoniae*, *A. baumannii*, *Pseudomonas aeruginosa*, and *Enterobacter* spp.) that exhibit resistance to multiple antibiotics [2]. The genomes of Acinetobacter strains are flexible and adaptive, and they tend to accumulate antibiotic resistance determinants, including mobile genetic elements, via horizontal gene transfer. To date, many *A. baumannii* strains have been isolated and sequenced.

The clustered regularly interspaced short palindromic repeats (CRISPR)-Cas system is an acquired immune system widely present in the genomes of archaea (~90%) and bacteria (~40%), which acts as a defense mechanism against the invasion of exogenous nucleic acids, such as phages and plasmids [3]. The details of how the CRISPR-Cas system provides immunity are shown in Figure 1. CRISPR-Cas system consists of the CRISPR arrays, leader sequences, and CRISPR-associated (cas) genes (Figure 1). CRISPR arrays consist of small direct repeats, which are separated by similarly sized spacers. There are 4–10 highly conserved *cas* genes encoding Cas proteins adjacent a CRISPR array [4]. Recent studies have shown that the CRISPR-Cas9 editing system can be used to target drug-resistant genes to reverse antibiotic resistance [5]; metastasis of the CRISPR array targeting chromosomal regions relevant to antibiotic resistance results in the death of the recipient bacteria [6]. In *A. baumannii*, the classification of the CRISPR-Cas system is generally type I-F [7].

With the extensive abuse of antibiotics in agriculture and public health in the past few decades, antibiotic-resistant bacteria emerged under evolutionary selection pressure. In a retrospective study of a grade A tertiary hospital in southern China, the resistance rate of *A. baumannii* to carbapenems increased significantly from 18% to 60% in 2012–2019, and the resistance to the antibiotics carbapenem, quinolone, and tetracycline was positively correlated with their usage [8].

Due to the rapid expansion of sequencing technology and its increased affordability, more whole-genome sequences (WGSs) of *A. baumannii* strains are being acquired. Bacterial de novo sequencing is an important tool for studying the evolutionary genetic mechanism and key functional genes of bacteria. It can be used for the identification of related genes of pathogenic bacteria, research on intraspecific evolution relationship, and genetic theory model biological research. In the contemporary era, bioinformatics analysis based on whole-genome sequencing has become a powerful approach to investigate the drug-resistance mechanism of *A. baumannii*.

From 2017 to 2021, we collected 400 *A. baumannii* strains from different hospitals of Jiangsu in China and screened for the presence of the CRISPR-Cas system by PCR; only strain AB43 carried a complete CRISPR-Cas system. Thus, AB43 strain was used to investigate the relationship between the CRISPR-Cas system and drug resistance. Our published study showed that the *csy1*-deleted mutant strain of AB43 was resistant to all of the tested antibiotics except Polymyxin B and colistin, and the wild type AB43 was sensitive to all of the antibiotics we tested, which indicated that the CRISPR-Cas system may be related to drug resistance [9]. Csy1, Cas proteins, and CRISPR RNA can form the Csy protein, and studies have shown that the Csy proteins strengthen target recognition [10]. Further research by our group has proved that the CRISPR-Cas system may target endogenous abaI boost antibiotic sensitivity of *A. baumannii* AB43 strain [11]. However, the genomic features of the *A. baumannii* AB43 strain were not studied. Analyzing the characteristics of the *A. baumannii* CRISPR-Cas-containing genome may provide valuable information in combating the drug resistance of this bacterium. In this study, we performed whole-genome sequencing and annotation of the AB43 strain to conduct a thorough analysis of the relationship between the CRISPR-Cas system and antibiotic resistance in this strain based on genomic characteristics.

## 2. Materials and Methods

### 2.1. Bacterial Strain

The clinical *A. baumannii* strain AB43 was isolated from the affiliated hospital of Yangzhou University in Jiangsu, China. The species was identified using Gram stain, 16S RNA sequencing, and the VITEK-2 system (bioMerieux, Marcy l’Etoile, France) in the clinical microbiology laboratory. *A. baumannii* was grown on Luria-Bertani (LB) agar or broth. The isolates were stored at −80 °C in LB broth containing 20% glycerol (*v*/*v*).

### 2.2. Genome Sequencing and Assembling

The genomic DNA of the *A. baumannii* strain AB43 was extracted with a bacterial whole genome extraction kit (TaKaRa, Dalian, China). The AB43 genome was sequenced using a PacBio RS II platform (Pacific Biosciences, Menlo Park, CA, USA) and Illumina HiSeq 4000 platform (Illumina, San Diego, CA, USA) at the Beijing Genomics Institute (BGI, Shenzhen, China). The PacBio platform uses four SMRT zero-mode waveguide sequencing arrays to generate subread sets. PacBio subreads (length < 1 kb) were removed. The program Pbdagcon (https://github.com/PacificBiosciences/pbdagcon, accessed on 28 May 2018) was used for self-correction to obtain highly reliable corrected reads. The high-quality draft genomic sequences obtained in the first two steps were preliminarily assembled using the Celera Assembler. Single-base corrections of the preliminary assembled results were performed using GATK (https://www.broadinstitute.org/gatk/, accessed on 30 May 2018) and the SOAP tool packages (SOAP2, SOAPsnp, SOAPindel). Self-alignment was performed on the assembly of sequences with high reliability, and end-to-end connection cyclization was performed at the position where there were overlapping terminal fragments. If multiple sequences were assembled, genomic sequences and plasmid sequences would be distinguished. To confirm the existence of any plasmid, the filtered Illumina reads were mapped using SOAP to the bacterial plasmid database (http://www.ebi.ac.uk/genomes/plasmid.html, accessed on 31 May 2018). The AB43 genomic sequences and plasmid sequences were submitted to NCBI GenBank.

### 2.3. Gene Prediction and Annotation

Gene prediction on the AB43 genome was performed by Glimmer3 (http://www.cbcb.umd.edu/software/glimmer/, accessed on 2 June 2018) with an interpolated Markov model (IMM) [12]. The tRNAscan-SE tool [13], RNAmmer software [14], and the Rfam database [15] were used to recognize non-coding RNA (tRNA, rRNA, and ncRNAs). PHAge Search Tool (PHAST) web server (http://phast.wishartlab.com/, accessed on 21 February 2022) was used to predict prophage regions [16]. Insertion sequences (ISs) were identified by ISFinder with the blast tool (https://www-is.biotoul.fr/blast/, accessed on 24 February 2022) [17].

The best hit was abstracted using the Basic Local Alignment Search Tool (BLAST) for function annotation. We used seven databases for general function annotation, including Kyoto Encyclopedia of Genes and Genomes (KEGG) [18], Clusters of Orthologous Genes (COG) [19], Gene Ontology (GO) [20,21], Translation of the EMBL database (TrEMBL) [22], evolutionary genealogy of genes: Non-supervised Orthologous Groups (eggNOG) [23], Non-Redundant Protein Database (NR) databases, and Swiss-Prot [22]. Resistance genes were identified based on the core dataset in the Comprehensive Antibiotic Resistance Database (CARD) [24].

The CRISPR-Cas system of AB43 was detected by CRISPRCasFinder(https://crisprcas.i2bc.paris-saclay.fr/CrisprCasFinder/Index, accessed on 18 June 2018) in the CRISPRCasdb, including repeat and spacer sequences [25]. The targets of spacer sequences were explored by taking advantage of the CRISPR Target (http://crispr.otago.ac.nz/CRISPRTarget/crispr_analysis.html, accessed on 2 March 2022) [26]. The genome data of reference strains used for circular comparison were downloaded from NCBI, including the *A. baumannii* strains ATCC19606 and ATCC17978.

### 2.4. Multilocus Sequence Typing

Multilocus sequence typing (MLST) of AB43 was searched from the PubMLST.org website on the basis of the Oxford and the Pasteur scheme (https://pubmlst.org/organisms/acinetobacter-baumannii, accessed on 3 March 2022) [27]. According to the results of whole-genome sequencing, seven housekeeping genes were analyzed, and a sequence type was obtained from the combination of seven loci.

### 2.5. Sequence Alignment and Phylogenetic Analysis

CRISPR-Cas-positive *A. baumannii* strains (AB0057, R2091, 5075UW, AB307-0294, ATCC BAA1605, CIP70.10, AC1633, AYE, D36, AB0057, A1, ATCC19606, 3207, A1429, 103, 104, 736,7835, AB3207, DETAB-P2, B8300, and AF401) from the CRISPR database (https://crispr.i2bc.paris-saclay.fr/crispr/, accessed on 12 March 2022) and AB43 were selected for multi-sequence alignment. Multiple sequence alignment of Cas1 and Cas3 amino acid sequences of *A. baumannii* were performed by DNAMAN version 9.0 (Lynnon Biosoft Company, San Ramon, CA, USA). The phylogenetic tree was constructed by the MEGA version 7.0 using the maximum likelihood method [28].

## 3. Results

### 3.1. Genome Sequence and General Features

The general characteristics of the AB43 genome are listed in Table 1. The whole-genome sequence of the *A. baumannii* strain AB43 was submitted to GenBank under accession number CP083181 (https://www.ncbi.nlm.nih.gov/nuccore/CP083181, accessed on 19 March 2022). A total of 175967 clean reads were generated with an average length of 8860 bp. The complete genome of AB43 consists of a 3,854,806 bp circular chromosome with 39.10% G + C content and a 104,309 bp circular plasmid (plasmid pAB43-1) with 39.99% G + C content. A total of 375 potential protein-encoding sequences were predicted in the AB43 genome. Compared with the genome size of two reference strains (*A. baumannii* ATCC 19606 and ATCC 17978), we found that AB43 has a slightly smaller gene number and chromosome size, but carries a larger plasmid (Figure 2, Appendix A). The sequence type of AB43 was ST705 (33-31-2-28-1-191-5) under the Oxford MLST scheme. With the Pasteur MLST scheme, the sequence type was ST132 (3-5-5-1-7-1-4), which belongs to CC132. Nine prophage sequences were found in AB43, six on the chromosome and three on the plasmid. One was intact on the chromosome, five were incomplete, and three were questionable (Appendix A).

### 3.2. CRISPR Array Analysis

One CRISPR array was found in AB43. A total of 106 repeat sequences and 105 spacers were collected from the confirmed CRISPR array (Figure 2). The 106 repeat sequences were composed of five different sequences, and the repeat sequences were 28 bp long with a sequence highly conserved. However, some nucleotide mutations were also observed in the repeat sequences (Figure 3). The length of spacer sequences was 27–34 bp, with an average of 32 bp. Spacers were all different and produced 105 unique spacer sequences. Since the spacer regions are derived from partially homologous invasive sequences, we analyzed all spacer sequences by searching for them in the bacteriophage genome database and the plasmid database. Based on the study of Palmer [29], above 90% identity (27 of 30 nucleotides) was regarded as significant. Under these criteria, identities to 28 of 105 *A. baumannii* AB43 CRISPR spacers were found. Among the 28 CRISPR spacers, 6 spacers were homologous in the GenBank-phage genome database and the other spacer sequences targeted to various plasmids (Appendix A). CRISPR spacer sequences of AB43 matched to many known mobile element sequences, such as phage integrase, phage tail protein, and recombinase family protein. Spacers matched to *VirB4* gene in plasmids may affect the virulence of the AB43 strain.

### 3.3. Sequence Alignment and Phylogenetic Tree Analysis

We selected 19 CRISPR-Cas-positive *A. baumannii* strains from the CRISPR database, which all had *cas* genes on their flanks. Multi-sequence alignment analysis showed that either the core protein Cas1 or the type-I-specific protein Cas3 show low identity between different CRISPR-Cas types (Figure 4), whereas the same subtypes of Cas1 and Cas3 are highly conserved (Figure 1). The protein sequence identity between type I-Fa Cas1 (ATCC19606 strain) and I-Fb Cas1 (AB43 strain) was 48.8%, and the protein sequence identity between I-Fa Cas3 (ATCC19606 strain) and I-Fb Cas3 (AB43 strain) was 28.64%. To better reveal *cas* genes in different CRISPR-Cas types across the *A.baumannii*, a phylogenetic tree of 20 *cas1* and *cas3* genes from published *A.baumannii* genomes based on aligned nucleotide sequences is shown in Figure 5. They could be roughly divided into two clusters on the phylogenetic tree (Figure 5A,B).

### 3.4. Genes Related to Antibiotic Resistance

A total of 22 antibiotic resistance genes were identified in the AB43 chromosome, while there is no antibiotic resistance gene in the plasmid (Table 2). There are two classes of β-lactamase among these resistance genes, namely, Class C: *bla*_ADC-156_, belonging to the ADC family cephalosporin-hydrolyzing, and Class D: *bla*_OXA-120_, belonging to the *bla*_OXA-51_ family. AB43 did not contain genes coding for class A and class B β-lactamases, both *bla*_ADC-156_ and *bla*_OXA-120_ genes lack insertion sequences upstream and downstream of them. In addition, AB43 harbored a fluoroquinolone resistance gene (*parC*), intrinsic peptide antibiotic resistance genes (*lpsB, lpxA, lpxC*), aminoglycoside-modifying enzyme gene *ant* (*3*″)*-IIc*, and efflux pump genes. Overexpression of efflux pumps is one of the important reasons for the drug resistance of *A. baumannii* [30]. The AB43 genome carries four major classes of efflux pump genes, including RND family, MFS family, SMR family, and MATE family, except for the ATP-binding cassette (ABC) transporters. Most of the ABC family of efflux pump genes is found in fungal and animal cells [31], with only a rare case of ABC family drug efflux pumps in Gram-negative bacteria.

### 3.5. Insertion Sequences

IS Finder was used to detect the ISs in the AB43 genome sequences. We identified five types of IS families and annotated 51 ISs in the AB43 chromosome (Table 3). There are 78.4% (40/51) of ISs belonging to *A. baumannii*, and the rest of the ISs are considered as derived from other Acinetobacter species. A total of 39.2% (20/51) of ISs belong to the IS3 family, and 39.2% (20/51) of ISs belong to the IS5 family. IS3 and IS5 are the main IS families existing in AB43.

## 4. Discussion

Several studies have suggested that technologies based on the CRISPR-Cas system and small molecules may pave the way for seeking advanced antibiotics antimicrobial treatments, which can reduce the harm caused by multidrug-resistant (MDR) pathogens. The CRISPR-Cas system also offers the opportunity to treat MDR infection by allowing the quantitative and selective elimination of a single bacterial strain based on sequence specificity [32,33].

In this study, we conducted a WGS of the *A. baumannii* strain AB43, which contains a type I-Fb CRISPR-Cas system and is sensitive to most antibiotics. To date, the CRISPR-Cas system has been divided into two main classes: class 1, including the type I, III, and IV systems, and class 2, including type II, V, and VI systems. In total, there are 33 different subtypes [34]. Kate et al. analyzed the WGS of most *A. baumannii* strains published on NCBI, and 15.7% of the strains carried the CRISPR-Cas system of type I-Fb, the most common subtype [35]. Cas proteins are encoded upstream of the CRISPR array and determine the system type [36,37]. Thus, multiple sequence alignments and phylogenetic tree analyses of *cas1* and *cas3* were examined in our study. All types of CRISPR-Cas systems have the *cas1* gene, but the *cas3* gene is a type I CRISPR-Cas system-specific marker. The Cas3 domains may promote similar ‘primed’ spacers integrated into the subtype I-F systems [38]. Compared with the I-Fb CRISPR-Cas system, the I-Fa CRISPR-Cas system lacks the *csy1* [36]. Alignment of Cas1 and Cas3 amino acid sequences of *A. baumannii* strains with different types of CRISPR-Cas systems showed low identity (Figure 4), and alignment of different subtypes alone showed high identity (Appendix A). This is consistent with the fact that the typing of the CRISPR-Cas system is based on the Cas protein sequences. According to the phylogenetic analysis, we found that *cas* genes were highly specific in different CRISPR-Cas subtypes but conservative in the same subtype. Our analysis showed that the sequence type of AB43 is ST132, based on the Pasteur protocol, and this ST belongs to CC132 that carries the *OXA-120* allele [2]. The presence of *OXA-120* gene in Table 2 also verified that the typing was correct.

Antimicrobial resistance can be intrinsic, which is an inherent characteristic of microorganisms that limits the action of antibacterial agents. In addition to intrinsic resistance and gene mutations, acquired resistance genes are an important genomic basis for antibiotic resistance, which are obtained through horizontal gene transfer mediated by mobile genetic elements (MGEs) [39]. Acquired resistance can also be transmitted between bacteria by horizontal gene transfer of antibiotic resistance genes (ARGs). The CRISPR-Cas system can resist the invasion of foreign substances [40]. The relationship between the CRISPR-Cas system and antibiotic resistance was diverse and there was a significant reverse relation in *enterococci* [29,41]; there was no prominent connection in *Escherichia coli* [42]. As the CRISPR-Cas system has the potential to limit the entry of mobile genetic elements, it was meaningful to investigate if there was a correlation between the existence of CRISPR-Cas and antibiotic resistance genes. Li’s research revealed that the CRISPR-Cas system of *Klebsiella pneumoniae* could interfere with the acquisition of exogenous plasmids or phages carrying antibiotic-resistant genes and maintain the sensitivity of strains to antibiotics [43]. Although our published study shows that *A. baumannii* AB43 was sensitive to most of the antibiotics, including gentamicin, doxycycline, minocycline, imipenem, kanamycin, ciprofloxacin, ceftriaxone, tigecycline, erythromycin, polymyxin B, colistin, and rifampin [9], the genome contains a variety of antibiotic resistance-associated genes. The number of antibiotic resistance genes in AB43 was not much lower than the “No CRISPR/No Cas” *A. baumannii* strain in the research of Tyumentseva [44]. These inconsistencies may be because the presence or absence of antibiotic resistance genes is not always predictive of the phenotype [45]. The reason that the CRISPR-Cas-containing *A. baumannii* strain is susceptible to antibiotics is perhaps that the CRISPR-Cas system efficiently limits the acquisition of acquired antibiotic resistance genes [44]. Oxacillin is the main carbapenemase of *A. baumannii*, in which *OXA-23* plays a leading role, and carbapenem resistance caused by β-lactamase, usually associated with upstream insertion elements, results in the overexpression of the β-lactamase gene [46]. However, there were no insertion sequences near either *bla_OXA-120_* or *bla_ADC-156_* of AB43, which might have kept their expression at a relatively lower level. Additionally, AB43 did not carry class A and class B β-lactamases, so this may be the reason AB43 is sensitive to carbapenems.

Analysis of CRISPR-Cas system spacers found that the spacers matched the phage protein (tail protein, pilot protein, HNH endonuclease, integrase, DNA polymerase I, glycosyl hydrolase 108) and plasmid protein (VirB4, Y-family DNA polymerase, recombinase family protein, DUF1173 family protein, DUF2800 domain-containing protein, DNA topoisomerase) sequences. Spacers matching phage genes found in AB43 imply that it has been subjected to a massive phage attack, thus providing immunity against invasive bacteriophage. Several studies also show that CRISPR-Cas systems have an important role in virulence regulation in multiple pathogenic bacteria [47]. VirB4, as a virulence protein of pathogenic bacteria, is a key component of VirB secretion apparatus and has high similarity with members of the type IV secretion system [48]. Two specific spacers matched to VirB4 of plasmids present in the AB43 CRISPR array may influence the virulence of this strain. The mechanism may be as follows: the transcription of the VirB4-specific spacer generates an antisense RNA that can knock down this gene in *A. baumannii*. However, more experiments are needed to prove this hypothesis.

In conclusion, the complete genome sequence of an ST132 *A. baumannii* strain, AB43, which contains the type I-Fb CRISPR-Cas system was analyzed. Although the CRISPR-Cas system was present in AB43, the number of antibiotic resistance genes was not lower than the “No CRISPR-Cas system” strain published in other studies [44]. Due to the sensitive phenotype of AB43, the detailed mechanism of how the CRISPR-Cas system regulates the drug resistance of *A. baumannii* is not yet clear. It may be because the CRISPR-Cas system inhibits drug-resistance gene expression via endogenous gene regulation. This work provides a reference for future studies of how the CRISPR-Cas system regulates the drug resistance of *A. baumannii*, and it offers valuable genome data for the study of *A. baumannii.*

## Figures and Tables

**Figure 1 molecules-27-05665-f001:**
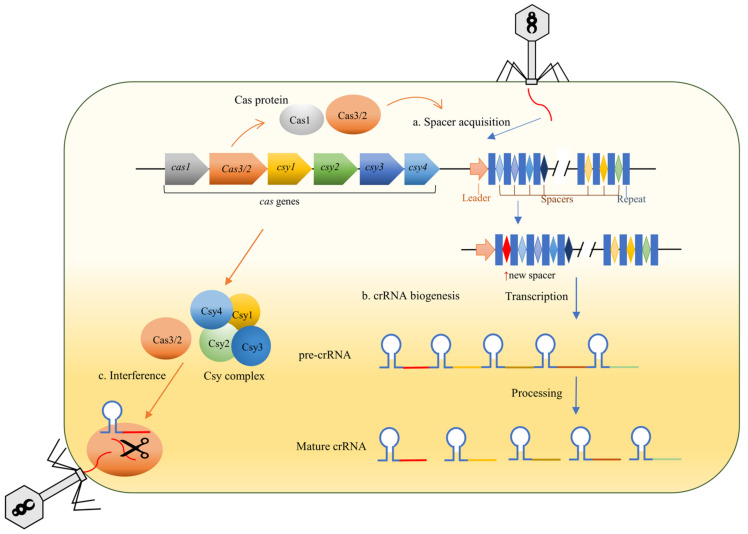
Schematic of how CRISPR-Cas system provides immunity. The immune function of type I-F CRISPR-Cas system can be roughly divided into three stages: adaptation (spacer acquisition), crRNA biogenesis, and target interference. When foreign genetic elements such as bacteriophage invade host bacteria, the relevant Cas proteins (Cas1 and Cas3/2) select small DNA fragments from foreign genetic elements and integrate them into the CRISPR array as the new spacer. The crRNA biogenesis begins with the transcription of CRISPR array into the pre-crRNA, which is then processed by specific nucleases into mature crRNA. In this process, Cas protein also begins to be expressed, allowing the recognition and binding of crRNA. In the final stage, if the phage invades again, the crRNA can be complementarily paired with the protospacer of the foreign nucleic acid, which guides the Cas3/2 nuclease-helicase to cut the target nucleic acid at a specific position, thereby resisting the invasion of the foreign genetic elements.

**Figure 2 molecules-27-05665-f002:**
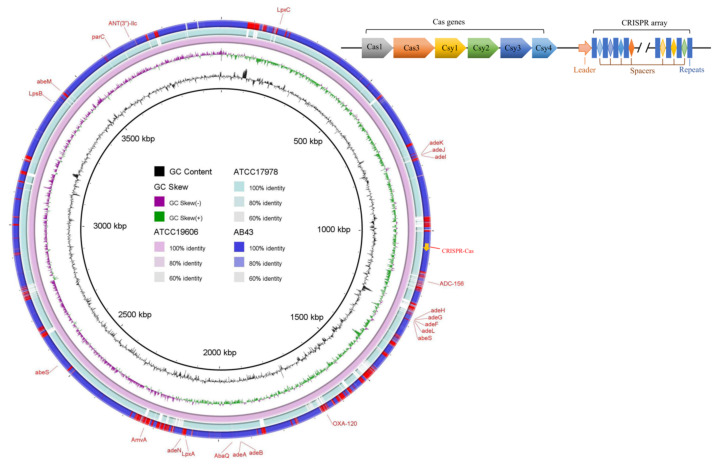
Basic information of the *Acinetobacter baumannii* strain AB43 chromosome. The ring diagram shows the comparison of three *A. baumannii* strain (AB43, ATCC19606, ATCC17978) chromosomes containing (or not containing) different CRISPR-Cas systems. AB43 contains type I-Fb CRISPR-Cas system, ATCC19606 contains type I-Fa CRISPR-Cas system, ATCC17978 did not contain CRISPR-Cas system. Rings from inside to outside: (1) the G + C content; (2) the GC skew; green and purple represent the GC skew (+) and GC skew (−), respectively; (3) *A. baumannii* ATCC19606 chromosome; (4) *A. baumannii* ATCC17978 chromosome; (5) clinical isolate *A. baumannii* AB43 chromosome. The genes associated with drug resistance and the CRISPR-Cas system in *A. baumannii* AB43 were marked in the figure. The AB43 CRISPR-Cas system is composed of *cas* genes, leader sequences, and CRISPR array. Type I system signature gene, *cas3*, is shown (orange). CRISPR arrays are composed of repeats (blue rectangle) and spacers (colored diamond). These repeats are separated by unique spacers.

**Figure 3 molecules-27-05665-f003:**
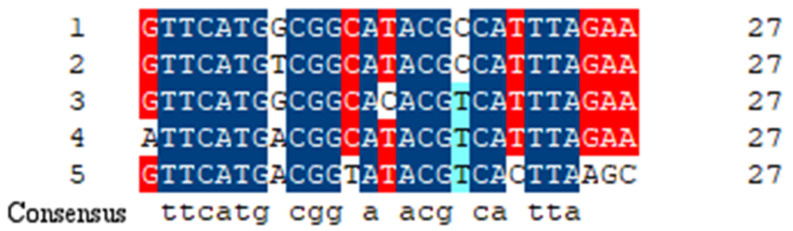
Nucleotide sequence alignment of the five different repeat sequences. The “consensus” line represents the same nucleotide.

**Figure 4 molecules-27-05665-f004:**
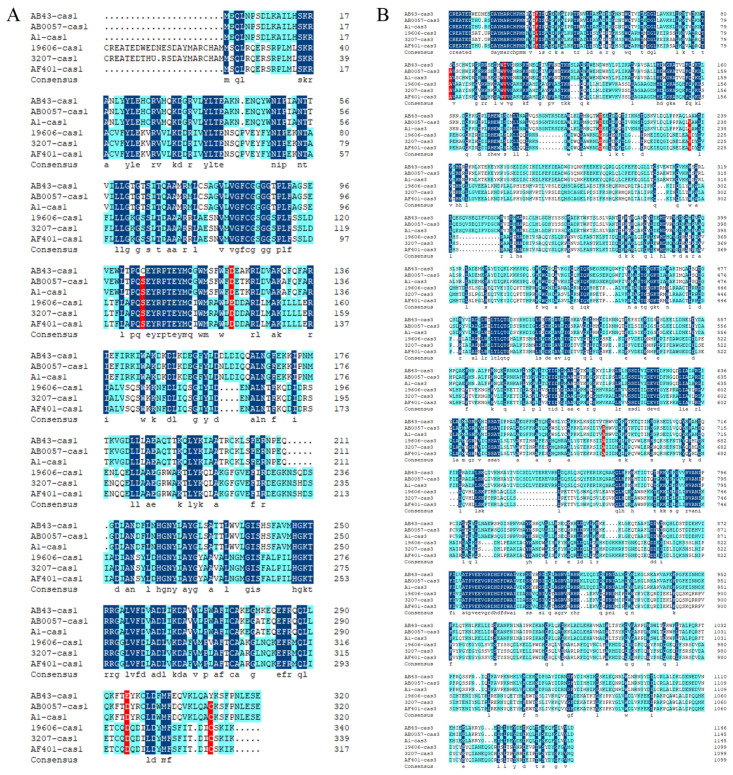
Alignment of Cas1 (**A**) and Cas3 (**B**) amino acid sequences of the *Acinetobacter baumannii* type I-Fb CRISPR-Cas system-containing strains (AB43, AB0057, R2091, 5075UW, AB307-0294, ATCC BAA1605, CIP70.10, AC1633, AYE, D36, AB0057, and A1) and type I-Fa CRISPR-Cas system-containing strains (ATCC19606, 3207, A1429, 103, 104, 736,7835, AB3207, DETAB-P2, B8300, and AF401). The “consensus” line represents the same amino acid with the reference sequence.

**Figure 5 molecules-27-05665-f005:**
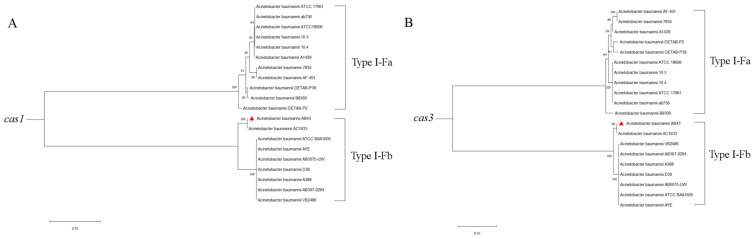
Phylogenetic tree of *cas1* (**A**) and *cas3* (**B**) genes, using the ClustalW algorithm to calculate and the maximum likelihood method to construct the phylogenetic tree in MEGA 7.0. The numbers at the top of the lines indicate the bootstrap value (1000 replications).

**Table 1 molecules-27-05665-t001:** General features of the *A. baumannii* strain AB43 genome.

Feature	Chromosome	Plasmid pAB43-1
Size (base pairs)	3,854,806	104,309
Topology	Circular	Circular
GC content (%)	39.10	39.99
No. of genes	3751	112
No. of protein-coding sequences	3487	88
No. of rRNA operons	18	-
No. of tRNA genes	74	-
No. of small ncRNA genes	4	-
No. of CRISPR	1	0
No. of prophage	6	3

**Table 2 molecules-27-05665-t002:** Antibiotic resistance genes in the *A. baumannii* strain AB43 genome.

AMR Gene Family	Coding Genes	Antibiotic Class	Locus Tags
Resistance-nodulation-cell division (RND) antibiotic efflux pump	*adeG*	Fluoroquinolone, tetracycline	AB43GL001197
	*adeB*	Glycylcycline, tetracycline	AB43GL001861
	*adeA*	Glycylcycline, tetracycline	AB43GL001862
	*adeN*	Macrolide antibiotic, fluoroquinolone antibiotic, lincosamide antibiotic, carbapenem, cephalosporin, tetracycline antibiotic, rifamycin antibiotic, diaminopyrimidine antibiotic, phenicol antibiotic, penem	AB43GL002039
	*adeK*	Macrolide, fluoroquinolone, lincosamide, carbapenem, cephalosporin, tetracycline, rifamycin, diaminopyrimidine, phenicol, penem	AB43GL000719
	*adeI*	Macrolide, fluoroquinolone, lincosamide, carbapenem, cephalosporin, tetracycline, rifamycin, diaminopyrimidine, phenicol, penem	AB43GL000721
	*adeJ*	Macrolide, fluoroquinolone, lincosamide, carbapenem, cephalosporin, tetracycline, rifamycin, diaminopyrimidine, phenicol, penem	AB43GL000720
	*adeH*	Fluoroquinolone, tetracycline	AB43GL001196
	*adeL*	Fluoroquinolone, tetracycline	AB43GL001199
	*adeF*	Fluoroquinolone, tetracycline	AB43GL001198
Major facilitator superfamily (MFS) antibiotic efflux pump	*AbaQ*	Fluoroquinolone antibiotic	AB43GL001891
	*AmvA*	Macrolide antibiotic, acridine dye, disinfecting agents and intercalating dyes	AB43GL002170
Small multidrug resistance (SMR) antibiotic efflux pump	*AbeS*	Fluoroquinolone	AB43GL001204
Multidrug and toxic compound extrusion (MATE) transporter	*AbeM*	Macrolide, acridine dye	AB43GL003308
Intrinsic peptide antibiotic-resistant Lps	*LpsB*	Peptide antibiotic	AB43GL003272
	*LpxA*	Peptide antibiotic	AB43GL002024
	*LpxC*	Peptide antibiotic	AB43GL000163
ANT(3″)	*ANT*(*3*″)*-Ⅱc* *Aminoglycoside*	Aminoglycoside	AB43GL003549
Fluoroquinolone-resistant parC	*parC*	Fluoroquinolone antibiotic	AB43GL003487
ADC beta-lactamases pending classification for carbapenemase activity	*ADC-156*	Cephalosporin	AB43GL001071
OXA beta-lactamase	*OXA-120*	Carbapenem, cephalosporin, penam	AB43GL001588

**Table 3 molecules-27-05665-t003:** Insertion sequences in the *A. baumannii* strain AB43 genome.

Sequences Producing Significant Alignments	IS Family	Group	Origin	Location (Start–End)
ISAba2	IS3	IS51	*Acinetobacter baumannii*	825271–826579, 889584–890892,
				1670545–1671853, 2234086–2235394,
				2260215–2261523, 2287668–2288976,
				3186375–3187683, 3188421–3189729
ISAba22	IS3	IS3	*Acinetobacter baumannii*	1767604–1768877, 1804912–1806185,
				1941275–1942548, 2083961–2085234,
				2253669–2254929, 2283894–2285167,
				2455194–2456467, 3276754–3278027
ISAcsp12	IS3	IS3	*Acinetobacter* sp.	90476–90852, 92164–92236,
				92308–92376, 3230992–3231229
ISAba5	IS5	IS903	*Acinetobacter haemolyticus*	828759–829798, 876433–877472;
				942334–943373, 1372703–1373742,
				1548861–1549880, 2276631–2277670
ISAba13	IS5	IS903	*Acinetobacter baumannii*	141776–142656, 271285–272323,
				1882841–1883879,1889761–1890799,
				1939734–1940660, 254414–255452
ISAba59	IS5	IS903	*Acinetobacter baumannii*	140737–141775, 1118270–1119308,
				1646101–1647139, 2091081–2092119,
				2289970–2291008, 2431933–2432971,
ISAba62	IS5	IS427	*Acinetobacter baumannii*	863210–863355, 863912–864022
ISAba64	IS256		*Acinetobacter baumannii*	3674955–3675184, 3765469–3765698
ISAba44	IS481		*Vibrio cholerae*	260755–260818
ISAcsp2	IS630		*Acinetobacter baumannii*	420808–421689, 696057–696938,
				1280457–1281338, 1528383–1529264,
				1846480–1847361, 2467893–2468774,
				2546510–2547391, 3009474–3010355

## Data Availability

Data available from the corresponding authors upon reasonable request.

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
