# Peer review of "Whole-Genome Analysis of Acinetobacter baumannii Strain AB43 Containing a Type I-Fb CRISPR-Cas System: Insights into the Relationship with Drug Resistance"

_molecules, 2022, doi:10.3390/molecules27175665_

Round 1

Reviewer 1 Report

Guo et al. has submitted a manuscript on the whole genome analysis of Acinetobacter baumannii and identified genes important for drug resistance. Overall, the genome analysis is well conducted and well written. I have the following suggestions for the authors to consider in their revised manuscript.

-Figure 5: Please mention how many bootstrap was used for the phylogenetic tree and the boostrap values at the branch nodes.  

-Mention the number of plasmid present, active and ambiguous prophage sequence number and CRISPR array number identified from the different types of Crispr system in this strain.

-Figure 2: Delete “This is a figure. Schemes follow another format. If there are multiple panels, they should 180 be listed as: (a) Description of what is contained in the first panel; (b) Description of what is contained in the second panel. Figures should be placed in the main text near to the first time they are cited. A caption on a single line should be centered.

Author Response

Dear reviewer,

Thanks for your kind considerations for the manuscript. We have substantially revised our manuscript after reading the comments. The followings are the point by point responses.

Comments and Suggestions for Authors

Overall, the genome analysis is well conducted and well written. I have the following suggestions for the authors to consider in their revised manuscript.

1)Figure 5: Please mention how many bootstrap was used for the phylogenetic tree and the boostrap values at the branch nodes.  

Response: Thank you for your suggestion. The number of bootstrap used for the phylogenetic tree and the boostrap values at the branch nodes were added in the Figure 5 legend. See lines 204-206.  

2)Mention thenumber of plasmid present, active and ambiguous prophage sequence number and CRISPR array number identified from the different types of Crispr system in this strain?

Response: Thank you for your suggestion. The number of plasmid present, active and ambiguous prophage sequence number and CRISPR array number were added in the revised manuscript. See lines 152,157 and 172.

3)Figure 2: Delete “This is a figure. Schemes follow another format. If there are multiple panels, they should 180 be listed as: (a) Description of what is contained in the first panel; (b) Description of what is contained in the second panel. Figures should be placed in the main text near to the first time they are cited. A caption on a single line should be centered.

Response: Thank you for your kindly reminder. We have deleted this description.

Reviewer 2 Report

Guo et al reported a Type I-Fb CRISPR-Cas system in a clinical A. baumannii isolate, compared and discussed CRISPR-Cas system from different A. baumannii strains. In general, this is a timely topic as antimicrobial resistance is one of the three greatest threats to human health globally and multidrug-resistant (MDR) A. baumannii is highlighted as a SERIOUS THREAT by the WHO. I have few comments on the presented work below. 

(1) From the title, the authors hightlighted to figure out the relationship between CRISPR-Cas system and drug resistance, however, its not clear in the result. I do not think the presented results sufficiently support this statement. Although, the previous study from the same group reported that the expression of csy1 "siginificantly decreased" in KO strain compared to WT and complementary strain. Since you Knocked out the gene, it obviously should not detect any expression of cys1. The author need to point out the experimental evidence of how CRISPR-Cas system contribute to AMR. 

(2) A ML tree is needed for cas1 and cas3 phylogenetic analysis, rather than NJ.

(3) Please carefully check the format through the whole manuscipt. (e.g., italic A. baumannii, gene names, lowe case )

Author Response

Dear reviewer,

Thanks for your kind considerations for the manuscript. We have substantially revised our manuscript after reading the comments. The followings are the point by point responses.

Comments and Suggestions for Authors

Guo et al reported a Type I-Fb CRISPR-Cas system in a clinical A. baumannii isolate, compared and discussed CRISPR-Cas system from different A. baumannii strains. In general, this is a timely topic as antimicrobial resistance is one of the three greatest threats to human health globally and multidrug-resistant (MDR) A. baumannii is highlighted as a SERIOUS THREAT by the WHO. I have few comments on the presented work below. 

(1) From the title, the authors hightlighted to figure out the relationship between CRISPR-Cas system and drug resistance, however, its not clear in the result. I do not think the presented results sufficiently support this statement. Although, the previous study from the same group reported that the expression of csy1 "siginificantly decreased" in KO strain compared to WT and complementary strain. Since you Knocked out the gene, it obviously should not detect any expression of cys1. The author need to point out the experimental evidence of how CRISPR-Cas system contribute to AMR. 

Response: Thank you for your suggestion. The previous study (The Involvement of the csy1 Gene in the Antimicrobial Resistance of Acinetobacter baumannii[1]) from our group proved that when csy1 gene of CRISPR-Cas system was knocked out, the bacterial resistance level increased. And another recently published study (CRISPR-Cas in Acinetobacter baumannii Contributes to Antibiotic Susceptibility by Targeting Endogenous AbaI[2]) by our group proved that the CRISPR-Cas system may target endogenous abaI boost bacterial antibiotic sensitivity.

Through WGS of the A. baumannii strain AB43, the genome contains a variety of antibiotic resistance–associated genes. Though  strain AB43 is susceptible to all antibiotics tested, the inconsistencies may be because the presence or absence of antibiotic resistance genes is not always predictive of the phenotype.This put forwards a new idea that the CRISPR-Cas system may inhibit drug-resistance gene expression via endogenous gene regulation, except to the published mechanism that the CRISPR-Cas system efficiently limits the acquisition of antibiotic resistance genes that maintain bacteria sensitive to antibiotics. The features (not multidrug-resistant or highly virulent) of strain AB43 are just what we needed, strains have complete CRISPR-Cas system are sensitive to antibiotics. Hence, overexpression the CRISPR-Cas system in the strain may restore the bacterial sensitive to antibiotics. This may offer a new strategy to combat drug-resistant A.baumannii.

(2) A ML tree is needed for cas1 and cas3 phylogenetic analysis, rather than NJ.

Response: Thank you for your suggestion. We have redone the phylogenetic analysis using the maximum likelihood method and replaced Figure 5.

(3) Please carefully check the format through the whole manuscipt. (e.g., italic A. baumannii, gene names, lowe case )

Response: Thank you for your kindly reminder. We have checked the format through the whole manuscript.

References

  1. Guo, T.; Sun, X.; Li, M.; Wang, Y.; Jiao, H.; Li, G., The Involvement of the csy1 Gene in the Antimicrobial Resistance of Acinetobacter baumannii. Frontiers in medicine 2022, 9, 797104.
  2. Wang, Y.; Yang, J.; Sun, X.; Li, M.; Zhang, P.; Zhu, Z.; Jiao, H.; Guo, T.; Li, G., CRISPR-Cas in Acinetobacter baumannii Contributes to Antibiotic Susceptibility by Targeting Endogenous AbaI. Microbiology spectrum 2022, e0082922.

Reviewer 3 Report

This manuscript by Tingting Guo et al. describes the relationship between drug resistance and the CRISPR-Cas system in A. baumannii.

The manuscript deserves many modifications before it is finally accepted for publication.

Prefer the passive turn of phrase.

Italicize the names of the bacteria.

Methods: The selection criteria for the strains analyzed should be detailed.

Methods: The software should be properly referenced and its version indicated.

Methods: How have the antibiotic susceptibility testing been performed and interpreted?

Results : WGS sequence must be deposited as indicated in resutls part, but indication must be indicated in methods.

Results : Table 2 must be placed in suppl. table more than in the core manuscript.

Results: Figure 4 and 5 are hardly readable .

Results .Figure 5. PLease indicated the bootstrap methodology.

Line 346-347 : delete these sentences.

IRB statement : I think that, because this bacteria is of human origins, consent has to be obtained

Author Response

Dear reviewer,

Thanks for your kind considerations for the manuscript. We have substantially revised our manuscript after reading the comments. The followings are the point by point responses.

Comments and Suggestions for Authors

The manuscript deserves many modifications before it is finally accepted for publication.

1)Prefer the passive turn of phrase. Italicize the names of the bacteria.

Response: Thank you for your kindly reminder. We regret such carelessness and these have been revised. 

2)Methods: The selection criteria for the strains analyzed should be detailed.

Response: Thank you for your kindly reminder. We have added the reasons for choosing this strain for analysis in the introduction section and marked in red. See lines 83-85.

3)Methods: The software should be properly referenced and its version indicated.

Response: Thank you for your kindly reminder. We checked and modified the software and its version.

4)Methods: How have the antibiotic susceptibility testing been performed and interpreted?

ResponseMinimum inhibitory concentrations (MICs) of antibiotics were used to determine the antimicrobial susceptibility of strain AB43. The standard broth microdilution assays were done according to the Clinical and Laboratory Standard Institute (CLSI) guideline[1]. Results were interpreted according to the CLSI guidelines[2]. The antimicrobial susceptibility testing (AST) on A.baumannii  was performed using the broth microdilution method. Firstly, antibiotics were dissolved in water to prepare a high-concentration storage solution. The solutions were filter sterilized and stored at −20℃. A. baumannii strain was subcultured 1:100 from an overnight culture, grown to an OD600 of 0.5, and then diluted 1:100 in CAMHB to obtain a cell density of approximately 1×106 CFU/ml. Antibiotics sulfate at various concentrations was prepared in CAMHB, and 50μl of each antibiotic concentration(0.125-512μg/ml) and 50μl of each bacterial culture to be tested were added to the wells of polystyrene 96-well microtiter plates and mixed thoroughly by pipetting. A positive control for growth consisting of 50μl of CAMHB and 50μl of bacterial culture and a sterility control consisting of 100μl CAMHB were also included. The MIC was defined as the lowest concentration that inhibited bacterial growth at 37 °C after incubation for 20-24h. According to the CLSI guidelines, the drug resistance standards of these drugs are as follows: doxycycline(≥16μg/ml), minocycline(≥16μg/ml), ceftriaxone(≥64μg/ml), imipenem(≥8μg/ml), gentamicin(≥16μg/ml), kanamycin(≥64μg/ml), ciprofloxacin(≥4μg/ml), polymyxin B(≥4μg/ml), and colistin(≥4μg/ml).

5)Results : WGS sequence must be deposited as indicated in resutls part, but indication must be indicated in methods.

ResponseThank you very much. The indication was indicated in methods and marked in red. See lines 116.

6)Results : Table 2 must be placed in suppl. table more than in the core manuscript.

Response: Thank you for your suggestion. We have placed Table2 in suppl. table.

7)Results: Figure 4 and 5 are hardly readable .

ResponseThank you very much. To provide significant and scientifically sound results10 gene variants of each type were used for comparison. The results were shown in Figure 4 and Figure S1. Figure legend of Figure 5 was rewritten.

8)Results .Figure 5. PLease indicated the bootstrap methodology.

Response: Thank you for your kindly reminder. The bootstrap methodology used for the phylogenetic tree were added in the figure legend. 

9)Line 346-347: delete these sentences.

Response: Line 346-347 were deleted.

10) IRB statement : I think that, because this bacteria is of human origins, consent has to be obtained.

Response: Thank you for your kindly reminder. IRB statement was added and marked in red. See lines 296-299.

References:

【1】Clinical and Laboratory Standards Institute. Methods for dilution antimicrobial susceptibility tests for bacteria that grow aerobically: Eleventh Edition M07. Wayne: CLSI; 2018.

【2】Clinical and Laboratory Standard Institute. Performance standard for antimicrobial susceptibility testing: twenty-eighth edition M100. Wayne: CLSI; 2018.

Reviewer 4 Report

Guo et al. describe the application of whole genome sequencing for analysis of A. baumannii strain containing a fully functional CRISPR-Cas system. In general, the manuscript is easy to comprehend. However, I did not find any reasons for describing this particular strain. To my opinion, the manuscript essentially lacks novelty and the analysis conducted was not scientifically sound.

Major comments:

1. I did not understand the point of the paper. The authors sequenced and analyzed only one strain possessing Crispr-Cas I-Fb system. This system was already described (e.g., 0.1371/journal.pone.0118205 – 2015). The strain did not have any remarkable features (e.g., it was not multidrug-resistant or highly virulent). The composition of spacers was not unique and did not provide remarkable insights into strain evolution.

2. The authors did not provide any proofs for their hypothesis of inhibiting drug-resistance gene expression via endogenous gene regulation. Only speculations were provided. The strain could lack resistance genes due to plenty of reasons, and no connection with the presence of CRIPR-Cas system was shown.

3. Only five strains were used for comparison purposes. Currently there are more than 5000 isolates with full genomes are available in public databases, and more than 100 of them carry CRISPR-Cas systems according to publications cited by authors.

4. Comparison of different cas genes based on one representative of each type cannot provide significant and scientifically sound results – at least 10 gene variants should be used for each type.

5. The accession code for the genome of AB43 strain in public databases should be provided. It must be available according to journal policies.

6. Efflux pumps like ade family genes are usually not specific to particular antibiotics and are not attributed to acquired resistance genes, so it is not clear why they are considered to be in connection with CRISPR-Cas system which, according to some publications, inhibits acquisition of acquired (mostly, plasmid or phage-mediated) genes.

Minor comments:

Please italicize cas genes (not proteins) throughout the manuscript

Line 19 – hyphen is not needed in ‘drug-resistance’

Line 24 – should be “are described”, not “is described”

Line 33 – should be “lower than in the…”

Figure 1 – I suggest providing a figure with increased resolution

Line 55 – should be either “systems consist” or “system consists”

Line 108 – please italicize A. baumannii

Line 134 – “best hit was abstracted” – does not make sense, please rephrase

Figure 2 – please remove template text from caption (starting from ‘This is a figure’ and ending with ‘should be centered’). Please italicize A. baumannii throughout the caption

Line 174 – should be “on a chromosome” because there is only one chromosome present

Line 185 – “or absence” here is not grammatically correct, please fix (e.g., ‘not containing’)

Line 186-187 – should be “Ab43 contains”, the same for other strains. In addition, please use “did not” since it is better pertinent to scientific language

Line 190 – genes cannot be ‘drug-resistant’. Please rephrase.

Line 193 – “The CRISPR array are composed+ - grammatically incorrect

Table 2 – I suggest moving this table to supplementary

Lines 227-228 – “into two branches on the phylogenetic tree branches” – please rephrase

Author Response

Dear reviewer,

Thanks for your kind considerations for the manuscript. We have substantially revised our manuscript after reading the comments. The followings are the point by point responses.

Comments and Suggestions for Authors

Guo et al. describe the application of whole genome sequencing for analysis of A. baumannii strain containing a fully functional CRISPR-Cas system. In general, the manuscript is easy to comprehend. However, I did not find any reasons for describing this particular strain. To my opinion, the manuscript essentially lacks novelty and the analysis conducted was not scientifically sound.

Major comments:

  1. I did not understand the point of the paper. The authors sequenced and analyzed only one strain possessing Crispr-Cas I-Fb system. This system was already described (e.g., 0.1371/journal.pone.0118205 – 2015). The strain did not have any remarkable features (e.g., it was not multidrug-resistant or highly virulent). The composition of spacers was not unique and did not provide remarkable insights into strain evolution.

ResponseThank you very much. The point of this paper is investigate if there is a correlation between the existence of CRISPR-Cas and antibiotic resistance genes due to that CRISPR-Cas system has the potential to limit entry of mobile genetic elements. Through WGS of the A. baumannii strain AB43, the genome contains a variety of antibiotic resistance–associated genes. These inconsistencies may be because the presence or absence of antibiotic resistance genes is not always predictive of the phenotype. Additionally, through WGS of the A. baumannii strain AB43, we may deploying the Type I CRISPR-Cas system as an antimicrobial to treat drug-resistant A.baumannii infection in the future, which is an attractive strategy compared with conventional antibiotic therapy. The features (not multidrug-resistant or highly virulent) of strain AB43 are just what we needed, strains have complete CRISPR-Cas system are sensitive to antibiotics. Hence, overexpression the CRISPR-Cas system in the strain may restore the bacterial sensitive to antibiotics.

From 2017-2021, we collected 400 A. baumannii strains from different hospitals of Jiangsu in China. And screened the presence of CRISPR-Cas system by PCR, only strain AB43 carried a complete CRISPR-Cas system. None of the other strains contained intact CRISPR-Cas system. Thus, AB43 strain was used to investigate the relationship of CRISPR-Cas system with drug resistance. In the future, if strain contained intact CRISPR-Cas system collected , we will continue sequencing and analysis. Although the presence of the CRISPR-Cas I-Fb system has been reported in other literature, strain AB43 having the different ST type and different features (not multidrug-resistant or highly virulent).

   Among the spacers, two specific spacers matched to VirB4 of plasmids present in the AB43 CRISPR array may influence the virulence of this strain. VirB4, as a virulence protein of pathogenic bacteria, is a key component of VirB secretion apparatus and has high similarity with members of the type IV secretion system. However, more experiments are needed to prove this hypothesis.

  1. The authors did not provide any proofs for their hypothesis of inhibiting drug-resistance gene expression via endogenous gene regulation. Only speculations were provided. The strain could lack resistance genes due to plenty of reasons, and no connection with the presence of CRIPR-Cas system was shown.

Response: Recently published research by our group proved that CRISPR-Cas system may target endogenous AbaI boost antibiotic sensitivity of A. baumannii AB43 strain[1].

Through the genomic characteristics of the A. baumannii strain AB3, which containing the Type I-Fb CRISPR-Cas system, we found that the number of antibiotic resistance genes was not lower than in the “no CRISPR-Cas system” strain. This put forwards a new idea that the CRISPR-Cas system may inhibit drug-resistance gene expression via endogenous gene regulation, except to the published mechanism that the CRISPR-Cas system efficiently limits the acquisition of antibiotic resistance genes that maintain bacteria sensitive to antibiotics.

  1. Only five strains were used for comparison purposes. Currently there are more than 5000 isolates with full genomes are available in public databases, and more than 100 of them carry CRISPR-Cas systems according to publications cited by authors.

Response: Thank you for your suggestion. Actually, we compared more than 20 strains carrying the CRISPR-Cas system, and the result is consistent with the result shown in Figure 4. For picture clarity and combined with the suggestion 4, we showed the comparison with 10 strains in this paper. See Figure 4 and Figure S1.

  1. Comparison of different cas genes based on one representative of each type cannot provide significant and scientifically sound results – at least 10 gene variants should be used for each type.

Response: We greatly agree with the reviewer's suggestion. 10 gene variants of each type were used for comparison. The results were shown in Figure 4 and Figure S1.

  1. The accession code for the genome of AB43 strain in public databases should be provided. It must be available according to journal policies.

Response: Thank you for your kindly reminder. The accession code was added in result 3.1(line 150).

  1. Efflux pumps like adefamily genes are usually not specific to particular antibiotics and are not attributed to acquired resistance genes, so it is not clear why they are considered to be in connection with CRISPR-Cas system which, according to some publications, inhibits acquisition of acquired (mostly, plasmid or phage-mediated) genes.

Response: We greatly agree with the reviewer's opinion. This research just want to show the presence of antibiotic resistance genes in strain AB43,and there is no resistance phenotype despite the presence of resistance genes. This put forwards a new standpoint that the CRISPR-Cas system may inhibit drug-resistance gene expression via endogenous gene regulation that maintain bacteria sensitive to antibiotics.

Minor comments:

1)Please italicize cas genes (not proteins) throughout the manuscript

Response: Cas genes were italicized throughout the manuscript.

2)Line 19 – hyphen is not needed in ‘drug-resistance’

Response: Hyphen was deleted.

3)Line 24 – should be “are described”, not “is described”

Response: Thank you for your kindly reminder. We revised the mistake.

4)Line 33 – should be “lower than in the…”

Response: “lower than in the…” was used in the manuscript.

5)Figure 1 – I suggest providing a figure with increased resolution

Response: Thank you for your suggestion. The resolution of Figure 1 was increased.

6)Line 55 – should be either “systems consist” or “system consists”

Response: “system consists” was used in the manuscript.

7)Line 108 – please italicize A. baumannii

Response: A. baumannii was italicized.

8)Line 134 – “best hit was abstracted” – does not make sense, please rephrase

Response: The sentence was rephrased.

9)Figure 2 – please remove template text from caption (starting from ‘This is a figure’ and ending with ‘should be centered’). Please italicize A. baumannii throughout the caption

Response: We are greatly sorry for making this mistake. The template text was removed.

10)Line 174 – should be “on a chromosome” because there is only one chromosome present

Response: Thank you for your kindly reminder. We revised the mistake.

11)Line 185 – “or absence” here is not grammatically correct, please fix (e.g., ‘not containing’)

Response: The sentence was rephrased.

12)Line 186-187 – should be “Ab43 contains”, the same for other strains. In addition, please use “did not” since it is better pertinent to scientific language

Response: The relevant sentences were rephrased.

13)Line 190 – genes cannot be ‘drug-resistant’. Please rephrase.

Response: The sentence was rephrased.

14)Line 193 – “The CRISPR array are composed+ - grammatically incorrect

Response: The sentence was rephrased.

15)Table 2 – I suggest moving this table to supplementary

Response: Thank you for your suggestion. We have placed Table2 in supplementary table.

16)Lines 227-228 – “into two branches on the phylogenetic tree branches” – please rephrase

Response: The sentence was rephrased and marked in red. See lines 194.

References

  1. Wang, Y.; Yang, J.; Sun, X.; Li, M.; Zhang, P.; Zhu, Z.; Jiao, H.; Guo, T.; Li, G., CRISPR-Cas in Acinetobacter baumannii Contributes to Antibiotic Susceptibility by Targeting Endogenous AbaI. Microbiology spectrum 2022, e0082922.

Round 2

Reviewer 2 Report

The authors had addressed all of my comments.

Reviewer 3 Report

Dear Authors, 

Most of my previous comments have been adressed, nevertheless, some minors remain. See thereafter.

1)Prefer the passive turn of phrase. Italicize the names of the bacteria.

Response: Thank you for your kindly reminder. We regret such carelessness and these have been revised.

-> Some passive turn of phrase remains to be done.

2)Methods: The selection criteria for the strains analyzed should be detailed.

Response: Thank you for your kindly reminder. We have added the reasons for choosing this strain for analysis in the introduction section and marked in red. See lines 83-85.

--> detail of the screening for CRISPRCas system must be indicated.

3)Methods: The software should be properly referenced and its version indicated.

Response: Thank you for your kindly reminder. We checked and modified the software and its version.

--> Suppress the website adress from the core manuscript and indicate it in the reference part of the manuscript. Some Database must be versionnized and referenced as  well.

5)Results : WGS sequence must be deposited as indicated in resutls part, but indication must be indicated in methods.

ResponseThank you very much. The indication was indicated in methods and marked in red. See lines 116.

--> Acecssion number must be indicated.

7)Results: Figure 4 and 5 are hardly readable .

ResponseThank you very much. To provide significant and scientifically sound results10 gene variants of each type were used for comparison. The results were shown in Figure 4 and Figure S1. Figure legend of Figure 5 was rewritten.

--> Figure 4 and Figure 5 remain non-readable (even after a 300%-zoom on Acrobat reader)

Reviewer 4 Report

I did not see any significant changes in the manuscript. As the authors state, they previously published the investigation of the strain with inactivated Cas system which was resistant to most antibiotics, and the current research is only a supplementary with a genomic sequence. No experiments were performed to prove the real activity of CRISPR-Cas system and the nature of antibiotic resistance. The comparisons made for cas genes are not statistically significant. I strongly suggest performing additional experiments or publish a purely genome descriptive paper in dedicated journal like Microbial Resource Announcements